# Systematic Review and Meta-Analysis on Burnout Owing to Perfectionism in Elite Athletes Based on the Multidimensional Perfectionism Scale (MPS) and Athlete Burnout Questionnaire (ABQ)

**DOI:** 10.3390/healthcare11101417

**Published:** 2023-05-13

**Authors:** Ji-Hye Yang, Hye-Jin Yang, Chulhwan Choi, Chul-Ho Bum

**Affiliations:** 1Department of Physical Education, Graduate School, Kyung Hee University, Seocheon-dong 1, Giheung-gu, Yongin-si 17104, Republic of Korea; didwlgp9@khu.ac.kr (J.-H.Y.); y0108577@khu.ac.kr (H.-J.Y.); 2Department of Physical Education, Gachon University, 1342 Seongnamdaero, Sujeong-gu, Seongnam-si 13120, Republic of Korea; 3Department of Golf Industry, College of Physical Education, Kyung Hee University, Seocheon-dong 1, Giheung-gu, Yongin-si 17104, Republic of Korea

**Keywords:** burnout, perfectionism, multidimensional perfectionism scale, athlete burnout questionnaire

## Abstract

Previous studies have shown that burnout negatively affects athletes’ mental health. To further explore this subject, we conducted a systematic review and meta-analysis by combining data from previous studies. This study followed the PRISMA guidelines for systematic and reliable research and completed data extraction using 10 databases and 8 keywords in December 2021. There were 93 cases of initially extracted data from the selected articles (*n* = 14) and the meta-analysis was conducted using the “meta” package, version 4.8-4 of R Studio 3.3.3, with data (k = 77) excluding other-oriented perfectionism data (k = 16). The results showed that self-oriented perfectionism had a negative effect on sports devaluation (SD) (ESr = −0.246, *p* < 0.001), and socially prescribed perfectionism had a positive effect on emotional/physical exhaustion (ESr = 0.150, *p* < 0.05) and SD (ESr = 0.138, *p* < 0.05). Furthermore, the test for publication bias showed that no groups had asymmetrical data, and four moderator analyses were conducted to prove the heterogeneity (*I*^2^) of the total effect size; however, there was no difference among groups (*Q_B_*), thereby resulting in unexplained variance. Consequently, this study presents variable data that determine the effects of perfectionism and burnout on elite athletes.

## 1. Introduction

### 1.1. Research Purpose and Significance

Perfectionism refers to self-conscious or obsessive attempts to achieve goals set by oneself or others [1]. Stoeber [2] views this as an individual personality factor. Perfectionism cannot be regarded as either good or bad because while it has negative effects, such as stress and depression [3], it also has positive effects, such as motivation for goal achievement [4].

Hewitt and Flett [5] developed the Multidimensional Perfectionism Scale (MPS) to identify perfectionism more clearly, representing it in three categories. The authors stated the following: self-oriented perfectionism (SOP) is setting goals on one’s own and performing them perfectly, socially prescribed perfectionism (SPP) is performing things perfectly owing to social and other pressure, and other-oriented perfectionism (OOP) is the perfectionist tendency to want others to satisfy one’s own perfect standards.

Studies of sports and athletes have often presented contradictory findings. Madigan et al. [6] discovered that excessive practice owing to perfectionism leads to injuries in athletes, and Sagar and Stoeber [7] revealed that perfectionism generates negative emotions, such as fear of poor performance or shame. In contrast, Curran et al. [8] found that perfectionism increases passion in athletes, and Ayas and Biçer [9] claimed that athletes with excellent performance are more likely to be perfectionists than others when it comes to achieving higher goals.

Based on these conflicting views, we focus on perfectionism from a negative perspective. Rotella et al. [10] argued that burnout in athletes is a state in which they are psychologically, emotionally, and physically exhausted, whereas Gustafsson et al. [11] claimed that the constant stress from pursuing perfectionism easily causes burnout symptoms in athletes. Considering its importance, scholars have conducted several related studies on burnout [12,13,14,15]. In addition, Madigan et al. [16] investigated the changing effects of perfectionism and burnout in long-term studies using repeated measures.

Raedeke and Smith [17] developed an Athlete Burnout Questionnaire (ABQ) to identify burnout among sports players. In this scale, burnout is measured based on a player’s decrease in achievement, mental/physical exhaustion, and devaluation. These questionnaires were translated into other languages, such as in Arce et al.’s [18] study, and have been used in previous research in various countries, such as Sharp et al.’s [19] study on Britons, to help prevent and measure burnout among sports players.

In previous studies, perfectionism has been measured using various tools, such as the MPS developed by Hewitt and Flett [5], the Multidimensional Inventory of Perfectionism in Sport (MIPS) developed by Stoeber et al. [20], and the Sport Multidimensional Perfectionism Scale 2 developed by Gotwals and Dunn [21]. The scales frequently used for burnout include the ABQ by Raedeke and Smith [17] and the Maslach Burnout Inventory (MBI) by Maslach and Jackson [22]. The above measures have been used less frequently in previous studies than the popular MPS and ABQ; however, the properties to be investigated are similar, indicating that the same sub-factors exist as in the MPS and ABQ.

These scales show conflicting results for the sub-factors. For example, Ho et al. [23] and Aghdasi [24], who used the MPS and ABQ, found negative results in the relationship between SOP and burnout; conversely, positive results were found in the relationship between SPP and burnout. Regarding OOP, which is the third factor of the MPS, Olsson et al. [25] and Park [26] found a positive relationship between OOP and burnout, whereas Zhang [27] and Kim et al. [28] found negative relationships, indicating that the results vary greatly among studies.

Therefore, this study aimed to determine the multidimensional relationship between perfectionism and burnout. Differences among the groups were analyzed using the same MPS and ABQ data from various scales for data consistency. Furthermore, this study provides meaningful data for improving athletes’ quality of life by discovering the relevance of burnout based on their perfectionist tendencies.

### 1.2. Research Question

The research questions are as follows.

What is the effect size of the impact on the types of burnout (emotional/physical exhaustion (E), reduced sense of accomplishment (R), and sports devaluation (SD)) depending on the type of perfectionism (SOP or SPP)?How does the effect size vary depending on the type of sport (individual, team, or individual/team sports)?How does the effect size vary depending on sex (male/both male and female)?How does the effect size vary depending on the nationality of participants (Asian/non-Asian)?How does effect size vary depending on the type of publication (academic journal or dissertation)?

## 2. Materials and Methods

### 2.1. Inclusion/Exclusion Criteria

The exclusion criteria were as follows: (1) studies that were not on sports; (2) studies that were not on elite athletes; (3) studies that were not full-text articles; (4) studies that did not examine the relationship between perfectionism and burnout; (5) studies with data that could not be used in this study; and (6) studies not written in English or Korean. The inclusion criteria were not included in the exclusion criteria and were based on papers suitable for this study. Moreover, this study followed the guidelines of the Preferred Reporting Items for Systematic Reviews and Meta-Analysis (PRISMA) for systematic research [29].

### 2.2. Literature Search

A literature search for this study was completed in December 2021, and 10 databases (“Google Scholar”, “ProQuest”, “PubMed”, “Scopus”, “Web of Science”, “Research Information Sharing Service” (RISS), “Korea Citation Index” (KCI), “Korean studies Information Service System”(KISS), “Kyobo Scholar”, “DBpia”) and 8 keywords (“Sports”, “Sport”, “Perfectionism”, “Perfect”, “Burnout”, “Exhaustion”, “Athletes”, “Athletic”) were used for a more comprehensive data search. The database and keyword selection for the data search were decided after discussion among all the authors.

### 2.3. Quality Assessment

To measure whether the selected articles were suitable for research, we used the “Quality Assessment and Validity Tool for Correlational Studies” [30], which is effective in assessing the quality of research. The assessment was classified into four components (design, sample, measurement, and statistical analysis). There are 13 total items, but 12 of them are 1 point, and the remaining item (“If a scale was used for measuring the dependent variable, was the internal consistency ≥ 0.70?”) is 2 points, adding up to 14 points. Articles scoring 1–4 points were considered low quality and were not suitable for research, whereas those scoring 5–9 points were considered of medium quality with moderate suitability. Those scoring 10–14 were considered high quality with excellent suitability for research.

The quality of the 14 articles included in this study was assessed as follows: For the dependent variables of the questions (“Was the dependent variable measured using a valid instrument?”, “If a scale was used to measure the dependent variable, was the internal consistency ≥ 0.70?”), the articles were assessed based on the standard of burnout, and all 14 articles were of high quality, with 10 points (*n* = 1), 11 points (*n* = 4), 12 points (*n* = 6), and 13 points (*n* = 3). The details are presented in Table 1.

### 2.4. Data Extraction

The data extracted in this study include all sports in the selected articles (e.g., taekwondo, judo, golf, rugby, soccer, and dancing), athletes’ average age and sex in each article (male, female, or both male and female), type of sports played by the athletes (individual sports, team sports, and both individual/team sports), athletes’ nationality (Asian/non-Asian), the perfectionism scale (e.g., MPS, and Performance Perfectionism Scale for Sport (PPS-S)) and burnout scale (e.g., ABQ and MBI) used in each article, and the type of publication (academic journals and dissertations). This study ultimately used data (k = 77) which excluded the OOP data (k = 16) from “r” (k = 93), which is the correlation value in the selected articles (*n* = 14).

### 2.5. Data Analysis

This study was conducted using the “meta” package, version 4.8-4 of R Studio 3.3.3, and the process of selecting data was determined through discussion with all authors. Statistics from the 14 articles that were ultimately selected were all Pearson correlation coefficients (r); Fisher’s Z was used to convert “r” to standardized effect size.

The data initially extracted from the 14 articles included 93 cases, but as the number was inadequate for OOP data (k = 16), publication bias could not be tested; thus, it was excluded from the total effect and moderator analysis. Therefore, 77 cases of data were used to determine the effect sizes of the SOP and SPP. The standard used to determine the total effect size of the meta-analysis was based on the effect size of Cohen [39] (p. 13), with a small effect of <0.1, a moderate effect of 0.3, and a large effect of >0.5. This indicates that the correlation between perfectionism and burnout can be understood depending on the direction of the effect size (positive or negative) and that the positive direction induces burnout even more, whereas the negative direction weakens burnout. Moreover, this study used the inverse of variances in each article and data and applied the random effects model considering the scale of unequal variables as well as the number of athletes for statistics.

Heterogeneity explains the variance in effect sizes. Thompson and Sharp [40] claimed that it is important to provide grounds for explaining heterogeneity when conducting a meta-analysis. Therefore, this study used the forest plot to explain the total effect size, confidence interval, and heterogeneity of all the data and explained heterogeneity with Q and *I*^2^ (%) among the multiple statistics that can measure heterogeneity (e.g., *I^2^, Q, τ^2^,* and *H^2^*). For *I^2^*, 40% or below indicates low heterogeneity, 30–60% indicates moderate heterogeneity, and 75–100% indicates considerable heterogeneity.

Four moderator analyses were conducted using a meta-ANOVA to explain the heterogeneity in this study. First, a moderator analysis of the subfactors was conducted using only MPS and ABQ data, excluding OOP data, as occurred in the total effect size analysis. Consequently, a moderator analysis was conducted using whole data (k = 77) from all the articles (*n* = 14), determining the effect size, which varied depending on the type of sport played by athletes (individual sports, team sports, and individual/team sports), sex (male/both male and female), nationality of athletes (Asian/non-Asian), and publication type (academic journals and dissertations).

Publication bias analysis was used to determine whether the selected data were biased. This study primarily confirmed visual asymmetry using an initial funnel plot and conducted an Egger’s regression test for statistical validation.

## 3. Results

### 3.1. Literature Search

We searched a total of 3871 articles using 10 databases and 8 keywords. Articles were excluded primarily because of redundancy (*n* = 215), as well as after reviewing titles and abstracts (*n* = 3423). Subsequently, an in-depth review was conducted based on the selection criteria; we ultimately selected 14 articles for the meta-analysis. The details are provided in the PRISMA for Systematic Reviews and Meta-Analyses flowchart (Figure 1).

#### Study Characteristics

Fourteen articles were ultimately selected for this study, and 77 cases of data were used from the 93 initial cases extracted. There were a total of 3496 athletes studied. Ho et al. [23] demonstrated correlation data of athletes that are hearing impaired (*n* = 209) and athletes that are not (*n* = 203); thus, the data of both groups were extracted.

Data extracted from the selected articles were all in values of “r”, and total effect size and moderator analyses were conducted with all included data. The MPS and ABQ were selected for moderator analyses related to the sub-factors because they were the most popular of all perfectionism and burnout scales. Of the total 14 articles used, 11 articles used both the MPS and the ABQ, 1 article used the PPS-S [41] derived from the MPS (e.g., Olsson et al. [25]), and 2 articles used the MBI with the same sub-variables as the ABQ [37,38].

There are three sub-factors (SOP, SPP, and OOP) of the MPS. Hewitt and Flett [5] claimed that those with SOP set high standards for themselves and make a ceaseless effort to meet those standards; those with SPP make an effort to meet the social standards or standards set by others, and those with OOP hope that others will meet the standards set by them. Moreover, there are three subfactors (E, R, and SD) in the ABQ, in which E indicates athletes’ physical and emotional exhaustion. Gustafsson et al. [42] argued that R is the reduced sense of accomplishment that athletes experience by not attaching much importance to improving their sports skills or making efforts and SD involves devaluing one’s abilities in sports and having negative thoughts that they might not be able to perform well. In addition, the MBI excludes SD and adds depersonalization.

The sports covered in the selected articles included team sports, such as handball, soccer, and baseball, as well as individual sports, such as track and field, swimming, dancing, taekwondo, and golf. The participants were athletes aged 11–55 years old. Details of the selected articles and moderator analyses are presented in Table 2.

### 3.2. Overall Effects of Perfectionism and Burnout

The total effect size for the data (k = 77) of the articles (*n* = 14) ultimately selected in this study was analyzed by converting “r” to Fisher’s *Z* and calculating down to three places of decimals. A random effects model was used because the selected articles included population sizes and scales that were not homogeneous. The effect size for the entire dataset (k = 77) was ESr = −0.003 (95% CI −0.066; 0.059, *p* = 0.91), and the heterogeneity was *I*^2^ =94% (*p* < 0.001), indicating that the negative effect between multidimensional perfectionism and burnout was not statistically significant; details are shown in Figure 2.

The average effect size of the meta-analysis of SOP and R out of the six groups (SOP and R group, SOP and E group, SOP and SD group, SPP and R group, SPP and E group, SPP and SD group) was ESr = −0.132 (95 % CI −0.234; −0.030, *p* = 0.13), the average effect size of SOP and E was ESr = −0.079 (95 % CI −0.192; −0.035, *p* = 0.17), and the average effect size of SOP and SD was ESr = −0.246 (95 % CI −0.344; −0.143, *p* < 0.001). For SPP, the average effect size of SPP and R was ESr = 0.126 (95 % CI −0.023; 0.270, *p* = 0.09), the average effect size of SPP and E was ESr = 0.150 (95 % CI 0.025; 0.271, *p* < 0.05), and the average effect size of SPP and SD was ESr = 0.138. Regarding the total effect size, only the SOP and SD group (*p* < 0.001), SPP and E group, and SPP and SD group (*p* < 0.05) showed statistically significant results. This was between a small effect (<0.1) and a moderate effect (0.3) according to Cohen’s (1998) [31] effect size standard; thus, it was smaller than moderate. SOP and SD (*I*^2^ = 86%, *p* < 0.001), SPP and E (*I*^2^ = 91%, *p* < 0.001), and SPP and SD (*I*^2^ = 88%, *p* < 0.001) also showed high heterogeneity.

Moreover, this meta-analysis used a funnel plot to verify publication bias, but it was difficult to visually identify data asymmetry; thus, Egger’s regression test was used for the second validation. There was no publication bias in any of the groups. Detailed results are provided in Table 3 and Figure 3, Figure 4, Figure 5, Figure 6, Figure 7 and Figure 8.

### 3.3. Results of Moderator Analyses

Four types of moderator analyses were used to prove the heterogeneity of this study and were conducted using the total data (k = 77) of all selected articles (*n* = 14). Moreover, since all moderator analyses conducted in this study were categorical variables, a meta-ANOVA was conducted on all of them: types of sports played by athletes (individual sports, team sports, and individual/team sports), sex (male/both male and female), nationality of athletes (Asian/non-Asian), and type of publication (academic journals and dissertations).

For the type of sports, individual sports were classified into golf, tennis, badminton, and boxing, and team sports into soccer, rugby, and handball. Appleton et al. [12] examined both individual and team sports at the same time on soccer, cricket, and tennis players. Regarding sex, there were no articles only on female athletes; thus, the participants were divided into two groups: only male or both male and female athletes. Data on the nationalities of the athletes were collected: Chinese and Korean athletes were classified as Asian, and English, British, and Turkish athletes were classified as non-Asian. For publication type, $I^{2}$ values could not be obtained from the SPP and R groups and the SPP and D groups owing to insufficient dissertation data. Thus, the results of the moderator analyses can be obtained from groups other than these two. Consequently, the $I^{2}$ of each group in the moderator analyses had some statistically significant values, lower than the heterogeneity of the total effect size (e.g., the individual and team sports group of the SOP and R group ($I^{2}=73\%\;$), the male athletes of the SOP and R group ($I^{2}$ = 67%), the non-Asian group of SOP and E group ($I^{2}$ = 75%), and the academic journals group of SOP and E group ($I^{2}$ = 81%)). However, the $Q_{B}$ of all the groups was not statistically significant, indicating that there was no difference between the groups. This implies that the heterogeneity was generated by an unexplained variance in the variance ratio among studies ($I^{2}$) which requires additional review in the future. Details are provided in Table 4.

## 4. Discussion

To more comprehensively and clearly identify the relationship between perfectionism and burnout in elite athletes, this study used not only the MPS and ABQ but also scales with the same sub-factors.

The relationship between these two variables has been frequently observed [33]. For example, Zhang et al. [43] found that burnout appears to have a negative effect on perfectionism among college students, and Mahmoudi-Shahrebabaki [44] identified a relationship between burnout and teachers. However, in sports, previous studies on elite athletes as well as their parents [11] and coaches [45] prove the importance of the relationship between perfectionism and burnout. For instance, Olsson et al. [25] found that a coach’s OOP perfectionist tendency caused player burnout. In addition, Appleton et al. [46] derived a perfectionist relationship between players and parents, and Seo and Kim [47] found that parental pressure had a positive effect on players’ perfectionist tendencies and burnouts.

However, few studies in sports academia have conducted systematic reviews and meta-analyses on the correlation between these two variables [48,49]. Moreover, they used different criteria for article selection, athletes, and moderator analyses. Thus, the results of this study are novel because few studies have conducted moderator analyses on all athletes, regardless of age, with a focus on MPS and ABQ.

The overall data effect size (ESr = −0.003, *p* = 0.91) between perfectionism and burnout derived from the results of this study was not statistically significant, which could be the result of the conflicting effects of SOP and SPP. This study showed that SOP decreased SD and SPP increased E and SD. Yu [50] discovered that perfectionism based on personal standards decreased exhaustion, whereas perfectionism demanded by parental expectations increased exhaustion. In these conflicting results, depending on the type of perfectionism, the key is in the athletes’ intentions to play sports. SOP is an effort to achieve the goal that individuals set for themselves and is not affected by others. According to Mouratidis and Michou [51], this motivates athletes to dedicate more time to practice. Curran et al. [8] claimed that SOP is found in athletes with strong will and passion. In contrast, SPP relates to being pressured by society or others for success one does not want. Gustafsson et al. [11] revealed that, even among athletes with similar perfectionism, those who were more pressured by others about their games showed a higher level of burnout.

It is not just perfectionism that shows different results depending on an individual’s will. Similar examples include harmonious passion and obsessive passion. Individuals with harmonious passion, which is similar to SOP, have a high self-identity because they are highly motivated to do things independently and are not greatly affected by others in life [52]. However, according to Schellenberg et al. [53], individuals with obsessive passion have low self-identity because they perform tasks regardless of their own will, such as SPP, and show many negative emotions, such as stress. Martínez-Alvarado et al. [54] examined the relationship between passion and burnout in young athletes and discovered that harmonious passion decreased burnout and obsessive passion increased burnout.

Therefore, the results of this study showing that SOP decreases SD and SPP increases E and SD are closely related to individual will. Moreover, in sports as well as in other competitive environments, perfectionism might have a significant effect on athletic performance; however, others’ perfectionism did not have a positive effect on continuously playing sports. Finally, the results of this study provide valuable data that will help elite athletes alleviate their burnout symptoms, depending on their type of perfectionism, and live an uninterrupted and healthy life as an athlete.

### Limitations and Future Directions

After systematically selecting articles to increase the validity and reliability of the study, we assessed the quality of the selected articles to derive results using unbiased data. While this study verified the significant relationship between perfectionism and burnout, it had a few limitations.

OOP occurs when an individual demands perfectionism from others. This kind of perfectionism may be found among teammates in team sports but is rarely found in individual sports. Therefore, there were insufficient data on OOP (not even enough for statistics) compared with that on SOP and SPP to conduct a meta-analysis, which is why data on OOP were excluded.

Moreover, this study used limited data on the same variables in the MPS and the ABQ. Therefore, future studies should include data from other scales to obtain more comprehensive results.

In the moderator analyses, no studies were conducted on only female athletes among the articles selected in this study, which is why only two groups (male, both male and female) were used; thus, in-depth analyses could not be conducted. In the moderator analysis for publication type, there were also insufficient dissertation data (k = 1) in the SPP and R groups and the SPP and SD groups; consequently, statistics could not be derived.

Although this study’s results concluded that the high heterogeneity of the total effect size was an unexplained variance, future studies should review this. Moreover, since perfectionism is not only found in elite athletes, conducting research that compares non-athletes with elite athletes is necessary. To obtain more in-depth and comprehensive results, data from various scales should be used to investigate the relationship between perfectionism and burnout.

## 5. Conclusions

We conducted a systematic review and meta-analysis to determine the multidimensional relationship between perfectionism and burnout in elite athletes. Of the six groups, SOP decreased SD, and SPP increased E and SD. For the test of publication bias, a funnel plot and Egger’s regression test were used to verify that there was no publication bias, both visually and statistically. For heterogeneity, all six groups had high heterogeneity (*I^2^* = 86–93%); thus, a total of four moderator analyses (type of sports played by the athletes, personality of the athletes, nationality of the athletes, and type of publication) were conducted. However, all moderator analyses showed that the differences among the groups were not statistically significant, indicating that the heterogeneity of the total effect size was an unexplained variance. In conclusion, this study presents valuable data on controversies regarding the ambivalence of perfectionism by revealing how SOP and SPP affect burnout symptoms based on a systematic analysis. This study will not only help elite athletes directly but also parents and coaches looking for strategies to alleviate burnout in athletes and help them have a satisfying sports career.

## Figures and Tables

**Figure 1 healthcare-11-01417-f001:**
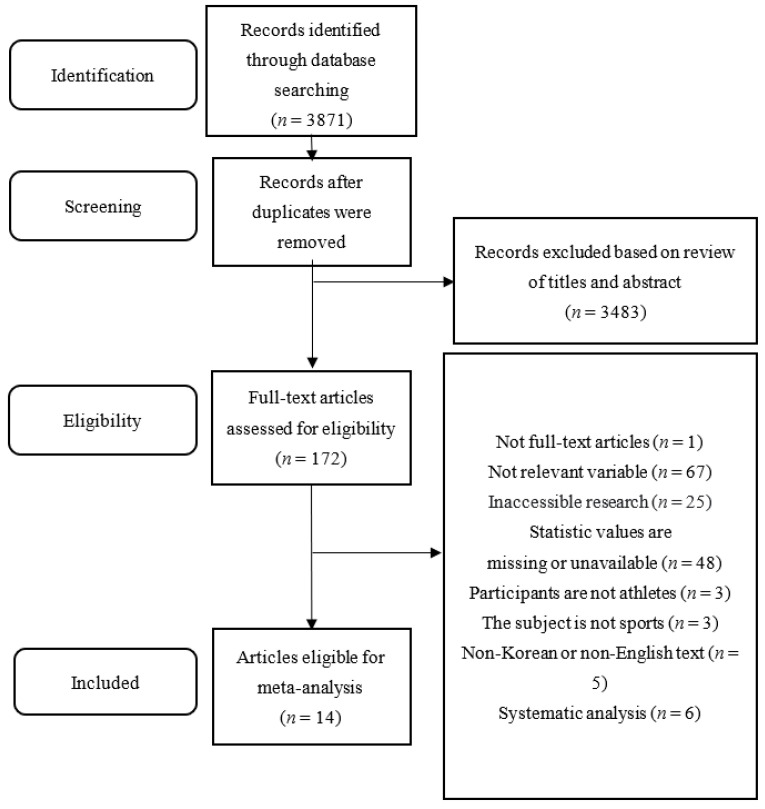
PRISMA flow chart of the literature analyzed in this study.

**Figure 2 healthcare-11-01417-f002:**
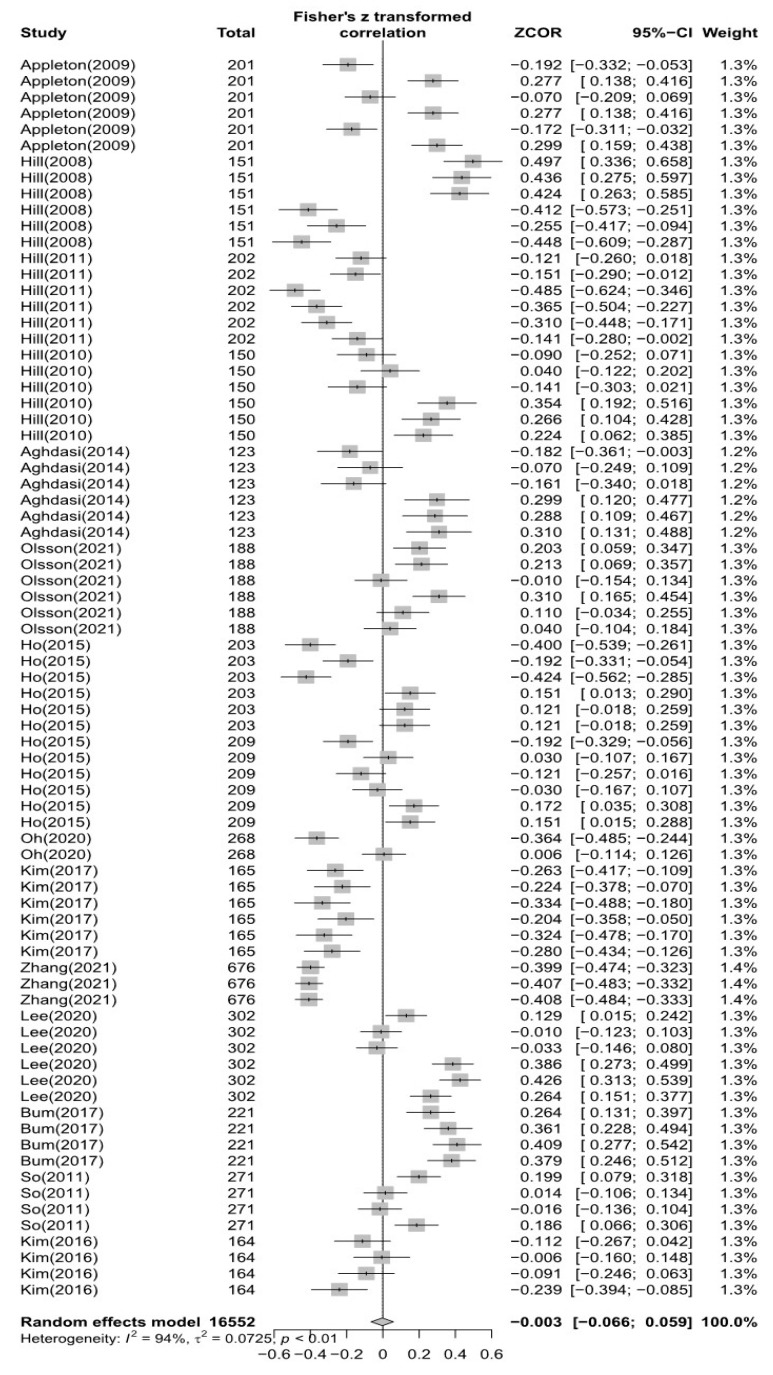
Forest plot of the total data [12,23,24,25,27,28,31,32,33,34,35,36,37,38].

**Figure 3 healthcare-11-01417-f003:**
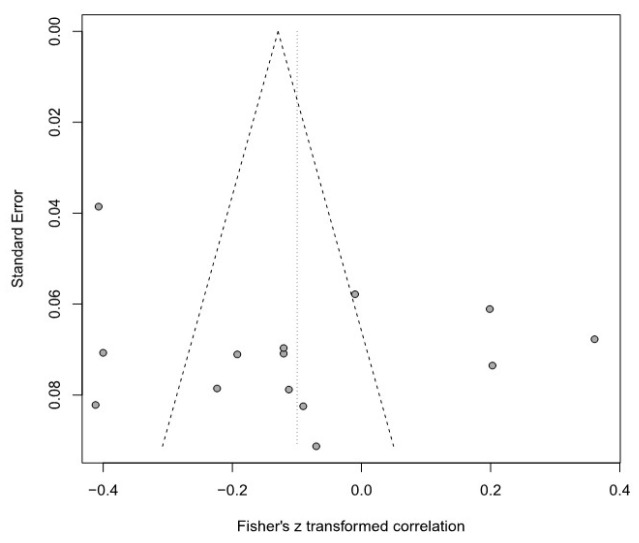
Funnel plot of the SOP and R group.

**Figure 4 healthcare-11-01417-f004:**
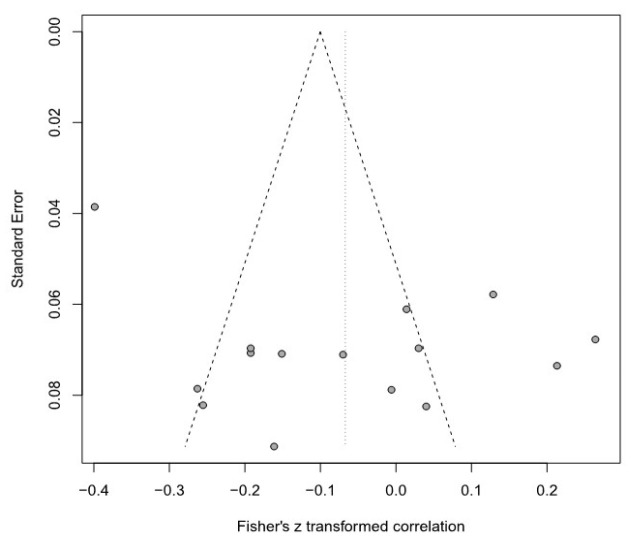
Funnel plot of the SOP and E group.

**Figure 5 healthcare-11-01417-f005:**
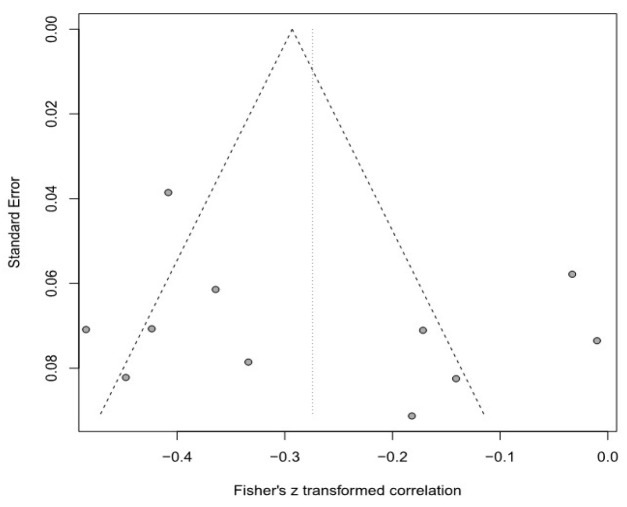
Funnel plot of the SOP and SD group.

**Figure 6 healthcare-11-01417-f006:**
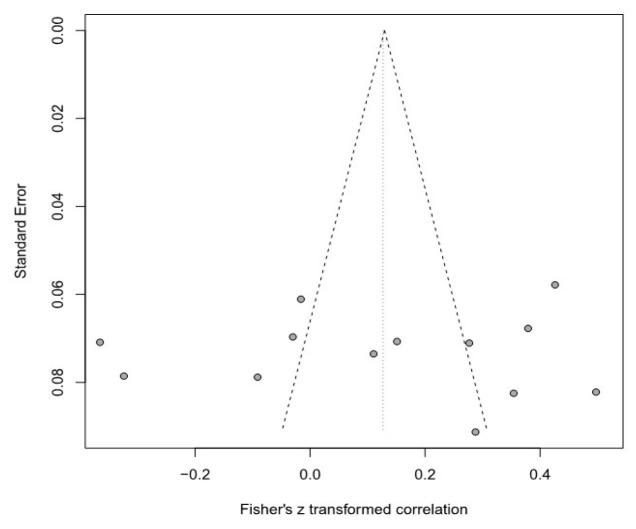
Funnel plot of the SPP and R group.

**Figure 7 healthcare-11-01417-f007:**
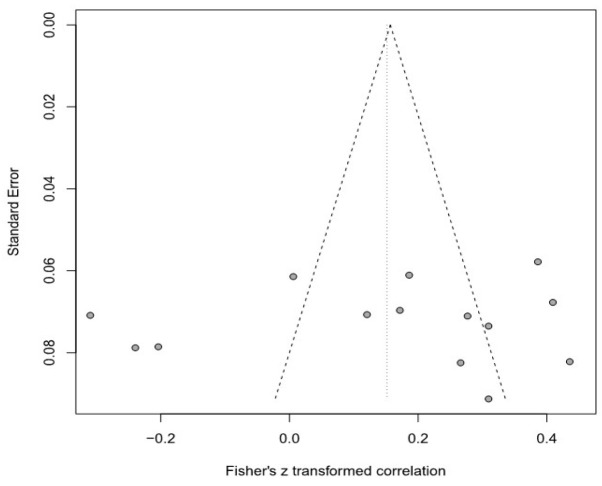
Funnel plot of the SPP and E group.

**Figure 8 healthcare-11-01417-f008:**
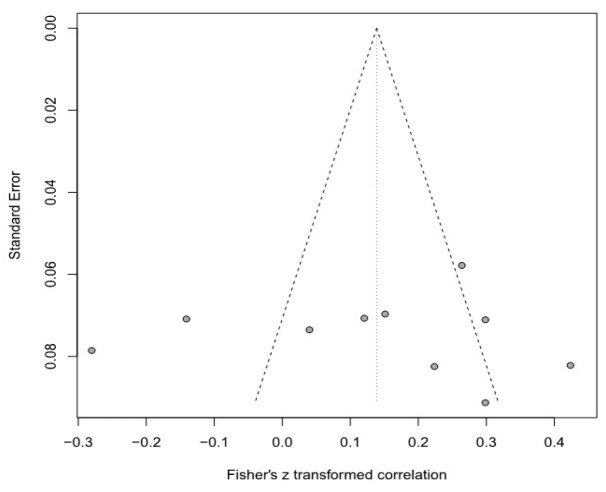
Funnel plot of the SPP and SD group.

**Table 1 healthcare-11-01417-t001:** Quality assessment and validity of the correlational studies.

First Author (Year)	Q1	Q2-1	Q2-2	Q2-3	Q2-4	Q2-5	Q3-1	Q3-2	Q4-1	Q4-2 (2 Points)	Q4-3	Q5-1	Q5-2	Outcome
Appleton (2009) [12]	Y	N/A	Y	Y	Y	N/A	Y	Y	Y	Y	Y	Y	Y	12(H)
Hill (2008) [31]	Y	N/A	Y	N	Y	N/A	Y	Y	Y	Y	Y	Y	Y	11(H)
Hill (2011) [32]	Y	N/A	Y	N	Y	N/A	Y	Y	Y	Y	Y	Y	Y	11(H)
Hill (2010) [33]	Y	N/A	Y	N	Y	N/A	Y	Y	Y	Y	Y	Y	Y	11(H)
Aghdasi (2014) [24]	Y	N	Y	N	Y	Y	Y	Y	Y	N/A	Y	Y	Y	10(H)
Olsson (2021) [25]	Y	N/A	Y	Y	Y	N/A	Y	Y	Y	Y	Y	Y	Y	12(H)
Ho (2015) [23]	Y	N/A	Y	Y	Y	N/A	Y	Y	Y	Y	Y	Y	Y	11(H)
Bum (2017) [34]	Y	N	Y	Y	Y	Y	Y	Y	Y	Y	Y	Y	Y	13(H)
So (2011) [35]	Y	N/A	Y	Y	Y	Y	Y	Y	Y	Y	Y	Y	Y	13(H)
Kim (2017) [36]	Y	N	Y	N	Y	N/A	Y	Y	Y	Y	Y	Y	Y	12(H)
Kim (2016) [28]	Y	N	Y	N	Y	Y	Y	Y	Y	Y	Y	Y	Y	12(H)
Zhang (2021) [27]	Y	N/A	Y	Y	Y	N/A	Y	Y	Y	Y	Y	Y	Y	12(H)
Oh (2020) [37]	Y	N	Y	N	Y	Y	Y	Y	Y	Y	Y	Y	Y	12(H)
Lee (2020) [38]	Y	N/A	Y	Y	Y	Y	Y	Y	Y	Y	Y	Y	Y	13(H)

Note. Y = Yes, N = No, N/A = No answer; Q1 = Is it a prospective study; Q2-1 = Probability sampling Q2-2 = Is the sample size justified; Q2-3 = Sample drawn from more than one site; Q2-4 = Is anonymity protected; Q2-5 = Response rate is more than 60%; Q3-1 = Is the outcome reliable; Q3-2 = Is the outcome measured using a valid instrument; Q4-1 = Is a dependent variable measured using a valid instrument; Q4-2 = If a scale was used to measure the dependent variable, is the internal consistency ≥ 0.70; Q4-3 = Is a theoretical framework used for guidance; Q5-1 = If multiple outcomes were studied, are correlations analyzed; Q5-2 = Are outliers managed; and H = High.

**Table 2 healthcare-11-01417-t002:** Summary of all the studies’ characteristics.

First Author (Year)	Participant Characteristics	Extracted Value	Perfectionism Questionnaires	Burnout Questionnaires	Type of Publication
Appleton (2009) [12]	(Individual, team) 201 junior elite male athletes aged 11–21 in England	Correlation (r)	MPS [5]	ABQ [17]	Article
Hill (2008) [31]	(Team) 151 male soccer players aged 10–18 in the UK	Correlation (r)	MPS [5]	ABQ [17]	Article
Hill (2011) [32]	(Team) 202 male rugby players aged 16–24 in the UK	Correlation (r)	MPS-H [5]	ABQ [17]	Article
Hill (2010) [33]	(Team) 150 male and female canoe/kayak athletes aged 13–55 in the UK	Correlation (r)	MPS [5]	ABQ [17]	Article
Aghdasi (2014) [24]	(Team) 123 male and female youth handball players in Turkey	Correlation (r)	MPS [5]	ABQ [17]	Article
Olsson (2021) [25]	(Individual) 190 male and female athletes with an average age of 20.54 in the UK	Correlation (r)	PPS-S [40]	ABQ [17]	Article
Ho (2015) [23]	(Individual, team) Male and female athletes in the UK Hearing impaired (*n* = 209), average age of 27.3 Not hearing impaired (*n* = 203), average age of 18.8	Correlation (r)	MPS [5]	ABQ [17]	Article
Bum (2017) [34]	(Individual) 221 male and female golf athletes from six universities in Korea	Correlation (r)	K-MPS [5]	MBI [22]	Article
So (2011) [35]	(Individual, team) 271 male and female athletes in middle and high schools in Korea	Correlation (r)	MPS [5]	MBI [22]	Article
Kim (2017) [36]	(Individual) 165 athletes in male and female business badminton teams in Korea as of 2016	Correlation (r)	MPS [5]	ABQ [17]	Article
Kim (2016) [28]	(Individual, team) 164 male athletes from one university in Korea	Correlation (r)	MPS [5]	MBI [22]	Article
Zhang (2021) [27]	(Individual) 676 male and female tennis players aged 19 and above in Korea and China	Correlation (r)	MPS [5]	ABQ [22]	Dissertation
Oh (2020) [37]	(Individual) 268 male and female athletes registered in the Boxing Association of Korea in 2019	Correlation (r)	MPS [5]	ABQ [17]	Dissertation
Lee (2020) [38]	(Individual) 302 male and female dancers in Korea	Correlation (r)	MPS [5]	ABQ [17]	Dissertation

**Table 3 healthcare-11-01417-t003:** Meta-analysis results of perfectionism and burnout.

	k	ESr	Confidence Interval	*I* ^2^	Egger’s (t)	Egger’s (df)	Egger’s (*p*)
1->3	k = 14	−0.132	−0.234; −0.030	93% ***	1.264	12	0.230
1->4	k = 15	−0.079	−0.192; −0.035	91% ***	1.935	13	0.074
1->5	k = 11	−0.246 ***	−0.344; −0.143	86% ***	0.982	9	0.351
2->3	k = 13	0.126	−0.023; 0.270	93% ***	−0.236	11	0.817
2->4	k = 14	0.150 *	0.025; 0.271	91% ***	−0.382	12	0.709
2->5	k = 10	0.138 *	0.009; 0.261	88% ***	−0.034	8	0.973

Note: 1, self-oriented perfectionism; 2, socially prescribed perfectionism; 3, reduced sense of accomplishment; 4, emotional/physical exhaustion; and 5, sports devaluation. *p* < 0.05 *, and *p* < 0.001 ***.

**Table 4 healthcare-11-01417-t004:** Results of the moderator analyses.

Moderator Analysis on the Type of Sports Played by Athletes											
SOP						SPP					
	k	ESr	*I* ^2^	*Q_B_*(df)	*p*-Value		k	ESr	*I* ^2^	*Q_B_*(df)	*p*-Value
1->7	k = 5	−0.016	91% **			4->7	k = 4	0.149	96% **		
1->8	k = 4	−0.137	97% **	0.81(2)	*p* = 0.66	4->8	k = 4	0.189	96% **	0.58(2)	*p* = 0.74
1->9	k = 5	−0.171	73% *			4->9	k = 5	0.058	78% **		
2->7	k = 6	−0.074	96% **			5->7	k = 5	0.181	93% **		
2->8	k = 4	−0.131	41%	0.29(2)	*p* = 0.86	5->8	k = 5	0.169	95% **	0.26(2)	*p* = 0.87
2->9	k = 5	−0.044	40%			5->9	k = 4	0.104	85% **		
3->7	k = 4	−0.196	93% **			6->7	k = 3	0.012	94% **		
3->8	k = 4	−0.196	82% **	0.71(2)	*p* = 0.70	6->8	k = 4	0.195	90% **	1.34(2)	*p* = 0.51
3->9	k = 3	−0.234	81% **			6->9	k = 3	0.187	44%		
**Moderator Analysis on the Athletes’ Sex**											
**SOP**						**SPP**					
	**k**	**ESr**	** *I* ^2^ **	***Q_B_*(df)**	***p*-Value**		**k**	**ESr**	$I^{2}$	$Q_{B}$ **(df)**	***p*-Value**
1->10	k = 4	−0.205	67% *	0.86(1)	*p* = 0.35	4->10	k = 4	0.077	96% **	0.17(1)	*p* = 0.67
1->11	k = 10	−0.062	95% **			4->11	k = 9	0.147	91% **		
2->10	k = 4	−0.119	45%	0.16(1)	*p* = 0.68	5->10	k = 4	0.039	96% **	1.24(1)	*p* = 0.26
2->11	k = 11	−0.064	94% **			5->11	k = 10	0.193	86% **		
3->10	k = 3	−0.351	82% **	1.60(1)	*p* = 0.20	6->10	k = 3	0.118	94% **	0.24(1)	*p* = 0.62
3->11	k = 8	−0.020	87% **			6->11	k = 7	0.116	85% **		
**Moderator Analysis on the Athletes‘ Nationality**											
**SOP**						**SPP**					
	**k**	**ESr**	$I^{2}$	$Q_{B}$ **(df)**	***p*-Value**		**k**	**ESr**	$I^{2}$	$Q_{B}$ **(df)**	***p*-Value**
1->12	k = 6	−0.033	96% **	0.74(1)	*p* = 0.39	4->12	k = 5	0.077	95% **	0.24(1)	*p* = 0.62
1->13	k = 8	−0.156	86% **			4->13	k = 8	0.157	92% **		
2->12	k = 7	−0.091	95% **	0.03(1)	*p* = 0.85	5->12	k = 6	0.094	94% **	0.56(1)	*p* = 0.45
2->13	k = 8	−0.068	75% **			5->13	k = 8	0.193	89% **		
3->12	k = 3	−0.351	82% **	1.60(1)	*p* = 0.20	6->12	k = 2	0.000	97% **	1.02(1)	*p* = 0.31
3->13	k = 8	−0.205	87% **			6->13	k = 8	0.173	82% **		
**Moderator Analysis on the Type of Publication**											
**SOP**						**SPP**					
	**k**	**ESr**	$I^{2}$	$Q_{B}$ **(df)**	***p*-Value**		**k**	**ESr**	$I^{2}$	$Q_{B}$ **(df)**	***p*-Value**
1->14	k = 12	−0.085	91% **	0.44(1)	*p* = 0.50	4->14	k = 12	0.101	92% **	1.38(1)	*p* = 0.23
1->15	k = 2	−0.208	97% **			4->15	k = 1	0.402	-		
2->14	k = 12	−0.044	81% **	1.66(1)	*p* = 0.19	5->14	k = 12	0.143	91% **	0.08(1)	*p* = 0.78
2->15	k = 3	−0.210	97% **			5->15	k = 2	0.194	95% **		
3->14	k = 9	−0.252	81% **	0.05(1)	*p* = 0.83	6->14	k = 9	0.123	88% **	0.38(1)	*p* = 0.53
3->15	K = 2	−0.221	97% **			6->15	k = 1	0.258	-		

Note. 1: Self-oriented perfectionism and a reduced sense of accomplishment; 2: self-oriented perfectionism and emotional/physical exhaustion; 3: self-oriented perfectionism and sports devaluation; 4: socially prescribed perfectionism and reduced sense of accomplishment group; 5: socially prescribed perfectionism and emotional/physical exhaustion group; 6: socially prescribed perfectionism and sports devaluation group; 7: individual sports; 8: team sports; 9: both individual and team sports; 10: male; 11: both male and female; 12: Asian nationality; 13: non-Asian nationality; 14: academic journal; 15: dissertation; and $Q_{B}$: between-group Q-value. *p* < 0.05 *, *p* < 0.01 **.

## Data Availability

Not applicable.

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
