# Peer review of "Systematic Review and Meta-Analysis on Burnout Owing to Perfectionism in Elite Athletes Based on the Multidimensional Perfectionism Scale (MPS) and Athlete Burnout Questionnaire (ABQ)"

_healthcare, 2023, doi:10.3390/healthcare11101417_

Round 1
Reviewer 1 Report
REVIEWER Comments:
This study presents a well-conducted systematic review and meta-analysis to explore the relationship between perfectionism and burnout in elite athletes. The authors have carefully examined the literature and selected relevant articles, ensuring the study's validity and reliability. By focusing on self-oriented perfectionism (SOP) and socially prescribed perfectionism (SPP), the authors have effectively demonstrated the contrasting effects of these two types of perfectionism on burnout in athletes. However, the authors need to check and correct some parts to ensure the completeness of the thesis.
Introduction
1. The introduction is well-structured, and the authors have provided a clear rationale for the study. However, it would be beneficial to further discuss the gaps in the existing literature and highlight the novelty of the current study.
2. Consider providing a brief overview of the Multidimensional Perfectionism Scale (MPS) and Athlete Burnout Questionnaire (ABQ) to familiarize readers with these tools.
3. Justify the choice of factors being investigated in the research questions and provide hypotheses based on existing literature.
Result
1. In the description of the study characteristics, it would be helpful to provide more information about the quality of the included studies, such as the study design, sample size, and potential biases. This information can help readers better assess the robustness of the meta-analysis findings.
2. Given the high heterogeneity observed in some of the overall effect sizes, it might be helpful to explore potential sources of this heterogeneity further. This could involve conducting additional moderator or sensitivity analyses to identify potential factors that may be driving the observed heterogeneity.
3. In addition to the Egger's regression test, the authors may consider providing a visual representation of the funnel plot for each group to enhance the presentation of the publication bias assessment.By addressing these areas, the Results and Discussion section will be more polished and easier to read, providing a more comprehensive understanding of the findings and their implications for coaching and athlete performance.
4. It would be helpful to include a clear explanation of the abbreviations used in the result tables. Providing a legend or a footnote that defines the abbreviations will improve the readability and understanding of the presented results.
Discussion:
1. The discussion section presents an insightful analysis of the study findings, highlighting the importance of individual will in the relationship between perfectionism and burnout. The connection between harmonious passion and obsessive passion is well-established, and the relevance to sports performance is clear. However, it would be beneficial to provide more examples or evidence from the literature to further support these claims. Additionally, it might be interesting to discuss the implications of the findings for sports psychology or coaching practices.
2. The authors have successfully provided an overview of the study's findings and their implications in the context of existing literature. It is commendable that they have identified the importance of individual will and the contrasting effects of self-oriented perfectionism (SOP) and socially prescribed perfectionism (SPP) on burnout.
3. As suggested earlier, it would be beneficial to further address the research question in the discussion section by reiterating the research question, discussing unexpected results, comparing the findings to previous studies, and suggesting future research directions.
Author Response
Review 1
This study presents a well-conducted systematic review and meta-analysis to explore the relationship between perfectionism and burnout in elite athletes. The authors have carefully examined the literature and selected relevant articles, ensuring the study's validity and reliability. By focusing on self-oriented perfectionism (SOP) and socially prescribed perfectionism (SPP), the authors have effectively demonstrated the contrasting effects of these two types of perfectionism on burnout in athletes. However, the authors need to check and correct some parts to ensure the completeness of the thesis.
Introduction
- Consider providing a brief overview of the Multidimensional Perfectionism Scale (MPS) and Athlete Burnout Questionnaire (ABQ) to familiarize readers with these → Based on your comment, the second paragraph of the Instruction has been modified.
- Justify the choice of factors being investigated in the research questions and provide hypotheses based on existing → The answer of this comment could be found in the section of 1.2. research question.
Result
- In the description of the study characteristics, it would be helpful to provide more information about the quality of the included studies, such as the study design, sample size, and potential biases. This information can help readers better assess the robustness of the meta-analysis → This has been described in 3.1.1. Study Characteristics.
- Given the high heterogeneity observed in some of the overall effect sizes, it might be helpful to explore potential sources of this heterogeneity This could involve conducting additional moderator or sensitivity analyses to identify potential factors that may be driving the observed heterogeneity. → We have already conducted four modulator analyses, and as a result, we could not explain the heterogeneity because there were no differences within the group.
- In addition to the Egger's regression test, the authors may consider providing a visual representation of the funnel plot for each group to enhance the presentation of the publication bias By addressing these areas, the Results and Discussion section will be more polished and easier to read, providing a more comprehensive understanding of the findings and their implications for coaching and athlete performance. → Figure 2 has been added.
- It would be helpful to include a clear explanation of the abbreviations used in the result Providing a legend or a footnote that defines the abbreviations will improve the readability and understanding of the presented results. → Please see the Note. under the tables.
Discussion
- The discussion section presents an insightful analysis of the study findings, highlighting the importance of individual will in the relationship between perfectionism and The connection between harmonious passion and obsessive passion is well-established, and the relevance to sports performance is clear. However, it would be beneficial to provide more examples or evidence from the literature to further support these claims. Additionally, it might be interesting to discuss the implications of the findings for sports psychology or coaching practices. → Additional information has been added in the second paragraph of the Discussion.
- The authors have successfully provided an overview of the study's findings and their implications in the context of existing It is commendable that they have identified the importance of individual will and the contrasting effects of self-oriented perfectionism (SOP) and socially prescribed perfectionism (SPP) on burnout. → Thank you for this comment.
- As suggested earlier, it would be beneficial to further address the research question in the discussion section by reiterating the research question, discussing unexpected results, comparing the findings to previous studies, and suggesting future research → Please see the 4.1. Limitations and Future Directions.
Reviewer 2 Report
Dear authors,
Thank you very much for your interesting article. I consider it is highly relevant and could be published in Healthcare.
Some considerations
GENERAL COMMENTS
Check if this journal admits two correspondence authors
An important aspect to consider are the articles that may have been published from 2021 to 2023. Therefore, it is necessary to carry out a new review so that this paper does not become outdated, including the latest research that meets the inclusion criteria.
INTRODUCTION
The introduction is adequate but perhaps quite summarized, although I think that what the authors want to express is clear.
MATERIALS AND METHODS
What were considered “professional elite athletes” in the studies?
The explanation of data analysis and data extraction is adequate
RESULTS
I think this is the best documented part of the study, congratulations. However, I consider too many articles were removed for missing values and inaccessible research. This must be indicated in limitations since it is probably a very high limitation. Why couldn't these articles be accessed? Did you try to contact the authors of all of them without receiving a response?
DISCUSSION AND CONCLUSIONS
Is right
Author Response
Review 2
Some considerations GENERAL COMMENTS
- Check if this journal admits two correspondence → Checked.
- An important aspect to consider are the articles that may have been published from 2021 to Therefore, it is necessary to carry out a new review so that this paper does not become outdated, including the latest research that meets the inclusion criteria. → This study was completed in late 2021.
Materials and methods
- What were considered “professional elite athletes” in the studies? The explanation of data analysis and data extraction is → The term "professional" has been deleted because it also includes general elite athletes, excluding professional athletes.
Results
- I think this is the best documented part of the study, congratulations. However, I consider too many articles were removed for missing values and inaccessible This must be indicated in limitations since it is probably a very high limitation. Why couldn't these articles be accessed? Did you try to contact the authors of all of them without receiving a response? → We tried to contact the authors of the paper applied to Statistical values are missing or unavailable as many times as possible to request statistics, but we did not receive any answer. No additional modifications were made in this paper.

Reviewer 3 Report
First of all, I would like to thank you for the opportunity to conduct this review. I would like to congratulate the authors for their work. However, there are numerous aspects that should be solved before being considered for publication.
Introduction
In the introduction I have a very important question: Why have the authors selected only the MPS and the ABQ? You yourselves previously indicate that the most used scales with MPS, MIPS, Sport-MPS-2, ABQ and MBO. Why do you limit yourselves only to these two scales to do the review and meta-analysis? This must be very well justified or it will constitute an important limitation to really know the relationship between burnout and perfectionism.
In the research question different types of burnout are mentioned, but nothing is mentioned in the introduction. The differences should be indicated and the reader should be provided with information about them.
Nothing was mentioned in the introduction about the differences in perfectionism and burnout between different sports modalities, between genders, nationalities, etc. In summary, the introduction is very poor and lacks information that really situates the reader with sufficient knowledge to understand what this manuscript contributes to the existing scientific literature.
The research aims have not been included.
Material and methods
In the inclusion and exclusion criteria. What do you mean by "studies with data that cannot be used in this study"? Please explain.
Also, you indicate the exclusion criteria, but where are the inclusion criteria?
Literature search
I would recommend including, at least as supplementary material, the search strategy used in each database and the results found in each of them.
Results
As for the results section, the funnel and forest charts are missing. You mention them, but the graphs, which are much more representative, do not appear.
Also, the article is supposed to be conducted on elite athletes, but Table 2 indicates that participants with 10-12 years were included. Is this really elite sport? Please justify this.
Discussion
I still see the same problem. Why haven't more scales been included to assess perfectionism and burnout, instead of limiting it to a single scale. This does not provide a true picture of the relationship between the two variables, only a limited view of what happens on these scales.
In addition, there is a lack of justification as to the reasons that may be giving rise to the relationships found between perfectionism and burnout. What is it that may be affecting the occurrence of these relationships? Are these reasons specific to the athletes, to the context? Further justification would be needed.
The inclusion of studies only in English and Korean is a limitation of the study.
Author Response
Review 3
First of all, I would like to thank you for the opportunity to conduct this review. I would like to congratulate the authors for their work. However, there are numerous aspects that should be solved before being considered for publication.
Introduction
- In the introduction I have a very important question: Why have the authors selected only the MPS and the ABQ? You yourselves previously indicate that the most used scales with MPS, MIPS, Sport-MPS-2, ABQ and Why do you limit yourselves only to these two scales to do the review and meta-analysis? This must be very well justified or it will constitute an important limitation to really know the relationship between burnout and perfectionism. → We totally agree with this comment. The sixth paragraph of the Introduction has been revised.
- In the research question different types of burnout are mentioned, but nothing is mentioned in the introduction. The differences should be indicated and the reader should be provided with information about → Based on your comment, the fifth paragraph of the Introduction has been added.
- Nothing was mentioned in the introduction about the differences in perfectionism and burnout between different sports modalities, between genders, nationalities, In summary, the introduction is very poor and lacks information that really situates the reader with sufficient knowledge to understand what this manuscript contributes to the existing scientific literature. → In the results of this study, there was no difference between sports events, gender, or countries, so there was no difference that could be mentioned in the introduction
- The research aims have not been → The research purpose has been stated in the last paragraph of the Introduction.
Methods
- In the inclusion and exclusion What do you mean by "studies with data that cannot be used in this study"? Please explain. → This means a paper using statistical data other than correlation statistical data.
- Also, you indicate the exclusion criteria, but where are the inclusion criteria? → Based on the comment, additional information has been added in 2.1. Inclusion/Exclusion Criteria.
Literature search
- I would recommend including, at least as supplementary material, the search strategy used in each database and the results found in each of → Literature Search has been described in Materials and Methods (2.2.) and Results (3.1.) respectively.
Results
- As for the results section, the funnel and forest charts are You mention them, but the graphs, which are much more representative, do not appear. → Figure 2 has been added.
- Also, the article is supposed to be conducted on elite athletes, but Table 2 indicates that participants with 10-12 years were included. Is this really elite sport? Please justify → No studies have adopted participants with 10-12 years as study subjects.
Discussion
- I still see the same Why haven't more scales been included to assess perfectionism and burnout, instead of limiting it to a single scale. This does not provide a true picture of the relationship between the two variables, only a limited view of what happens on these scales. → Based on this comment, the first paragraph of the Discussion has been revised.
- In addition, there is a lack of justification as to the reasons that may be giving rise to the relationships found between perfectionism and What is it that may be affecting the occurrence of these relationships? Are these reasons specific to the athletes, to the context? Further justification would be needed. → Additional information has been added in the second paragraph of the Discussion.
- The inclusion of studies only in English and Korean is a limitation of the → When this study began, papers that were not written in English or Korean were excluded from the exclusion criteria. The failure to identify research papers in all languages is generally not considered a research limitation.

Round 2
Reviewer 1 Report
Dear Author(s),
I have carefully reviewed the revised version of your manuscript, and I am pleased to see that the issues raised in my initial review have been adequately addressed. The thesis has been significantly improved, with clear editing and refinement of the content. I appreciate the efforts you have put into implementing the suggested changes and ensuring a high-quality presentation of your research.
Once again, thank you for your diligent work in addressing the concerns and improving the manuscript. I look forward to seeing the final version of your thesis published.
Reviewer 2 Report
Thank you
Reviewer 3 Report
The authors have provided answers to all the aspects requested.